# Improved GNSS Ambiguity Fast Estimation Reduction Algorithm

**DOI:** 10.3390/s23208568

**Published:** 2023-10-18

**Authors:** Xinzhong Li, Yongliang Xiong, Weiwei Chen, Shaoguang Xu, Rui Zhang

**Affiliations:** 1Faculty of Geosciences and Environmental Engineering, Southwest Jiaotong University, Chengdu 611756, China; lixinzhong@my.swjtu.edu.cn (X.L.); shaoguangxu@home.swjtu.edu.cn (S.X.); zhangry@my.swjtu.edu.cn (R.Z.); 2Department of Civil Engineering, Yibin Campus, Chengdu Technological University, Yibin 644000, China; cwwei1@cdtu.edu.cn

**Keywords:** GNSS, integer least squares, integer ambiguity, LLL reduction, partial size reduction

## Abstract

The fast and accurate solution of integer ambiguity is the key to achieve GNSS high-precision positioning. Based on the lattice theory of high-dimensional ambiguity solving, the reduction time consumption is much larger than the search time consumption, and it is especially important to improve the efficiency of the lattice basis reduction algorithm. The Householder QR decomposition with minimal column pivoting is utilized to pre-sort the basis vectors and reduce the number of basis vector exchanges during the reduction process by partial size reduction and relaxing the basis vector exchange condition to improve the reduction efficiency of the LLL algorithm. The improved algorithm is validated using simulated and measured data, respectively, and the performance advantages and disadvantages of the improved algorithm are evaluated from the perspectives of the extent of reduction basis orthogonality and the quality of reduction basis size reduction. The results show that the improved LLL algorithm can significantly reduce the number of basis vector exchanges and the reduction time consumption. The HSLLL and PSLLL algorithms with the Siegel condition as the basis vector exchange condition have a better reduction effect, but are slightly less stable. The PLLLR algorithm significantly improves the search ambiguity resolution efficiency, which is conducive to the rapid realization of ambiguity resolution.

## 1. Introduction

The resolution of integer ambiguity has a significant impact on the high-precision navigation and positioning results of the carrier phase. The LAMBDA method is currently recognized as the most theoretically rigorous, most efficiently solved, and most widely used ambiguity solving method [1]. It is based on the integer least squares model and reduces the search space by reducing the correlation between the variance components of the ambiguity in order to improve the search efficiency [2]. In addition, many scholars have carried out a lot of fruitful research on the decorrelation algorithm: Liu et al. proposed an approach to united ambiguity decorrelation from the perspective of LU decomposition [3]; Xu proposed a decorrelation algorithm for inverse integer Cholesky decomposition using a pre-sorting strategy [4,5]; Zhou proposed the (inverse) paired Cholesky integer transformation algorithm using upper and lower triangular Cholesky decomposition [6,7]; and Chang et al. improved the LAMBDA algorithm by using a greedy algorithm and lazy transformation strategy, and proposed the MLAMBDA algorithm [8].

Ambiguity resolution is an integer least-squares problem, which is essentially equivalent to the closest vector problem (CVP) in lattice theory, which is an NP-hard problem [9,10]. In order to obtain the nearest vector, it is usually necessary to reduce the basis vector. That is, the integer Gaussian transform is utilized to reduce the correlation of the basis vector, and the basis vector is sorted according to a certain criterion in order to obtain the shortest possible reduced basis. Among them, the LLL reduction algorithm is the most popular [11]. Therefore, the related methods of ambiguity resolution can be placed in the framework of lattice theory, thus injecting new vitality into the in-depth study of integer ambiguity resolution. Hassib et al. first introduced the LLL reduction algorithm to GNSS ambiguity resolution [12]. Grafarend introduced the solution principle of the LLL algorithm and carried out a detailed data analysis [13]. L.Z. Lou proposed to improve the original LLL algorithm by using a new judgment criterion in response to the defect of iterative non-convergence in the LLL algorithm [14]. Since the LLL algorithm introduces a large rounding error during the rounding process, Z.P. Liu et al. proposed an improved LLL algorithm based on overall matrix rounding [15]. R.H. Yang et al. improved the LLL algorithm by reordering the Gram–Schmidt orthogonal basis [16]. L. Fan and K. Xie improved the LLL algorithm from the perspective of reducing the length of the reduction basis vector [17,18]. Jazaeri et al. compared and analyzed the performance differences between the LAMBDA algorithm and the LLL algorithm [19]. Ling and Howgrave-Graham pointed out that the core of the LLL algorithm lies in the basis vector exchange by analyzing the characteristics of the size reduction and basis vector exchange in the LLL algorithm, and based on this, they proposed the ELLL (Effective LLL) algorithm with partial size reduction [20]. Xie et al. analyzed the effectiveness of the size reduction in the LLL algorithm and proposed the PLLL (Partial LLL) algorithm that selectively performs the size reduction of a column vector in response to the large truncation error of the ELLL algorithm [21]. L.G. Lu et al. improved the LLL algorithm using the greedy selection of the basis vector and partial column vector reduction to reduce the computational complexity of the LLL algorithm [22]. H. Lv et al. improved the original LLL algorithm by using delayed size reduction and partial size reduction in order to reduce the redundant size reduction during the reduction process [23]. Li et al. improved the LLL algorithm based on the Householder transform by using a symmetric pivoting strategy [24].

At the same time, relevant scholars have also conducted in-depth research on the correlation between the LAMBDA decorrelation algorithm and LLL reduction algorithm in lattice theory, as well as the performance evaluation index of lattice basis reduction assisting ambiguity resolution. J.N. Liu et al. theoretically proved the equivalence between the decorrelation and the lattice basis reduction [9]. Lannes further proved the equivalence of the LAMBDA decorrelation algorithm and the LLL algorithm [25]. Borno et al. pointed out through theoretical analysis that simple integer Gaussian transformations do not affect the efficiency of the search for integer ambiguity [26]. Jazaeri et al. analyzed the relationship between commonly used reduction performance evaluation indexes (condition number and orthogonal defect) and search efficiency, and pointed out that the above indexes could not accurately measure the efficiency of an accelerated ambiguity search for lattice basis reduction methods [27]. L.G. Lu et al. compared and analyzed the performance of the LAMBDA decorrelation algorithm and the LLL reduction algorithm under different decomposition methods. They categorized and generalized the common decorrelation (reduction) evaluation indexes from a geometric perspective. It is further illustrated that different evaluation indexes are not directly related to the search efficiency of ambiguity [28].

In view of this, this paper proposes new improved algorithms on the basis of LLL and PLLL algorithms for the characteristics of ambiguity resolution, and appropriately relaxes the exchange conditions of the basis vector in order to reduce the reduction time consumption and improve the computational efficiency of ambiguity resolution. The effectiveness of the improved algorithms and the reduction performance are verified by simulation and measured data.

## 2. Methods and Improvement Strategies

The following describes the notations to be used in this paper. The set of real and integer formed by n-dimensional vectors are denoted by ℝn and ℤn, respectively. MATLAB notation is used to represent submatrices. Specifically, if A=ai,j∈ℝ, then Ai,: denotes the i-th row, A:,j denotes the j-th column, and Ai1:i2,j1:j2 the submatrix formed by rows i1 to i2 and columns j1 to j2. For the element i,j of A, it is denoteb by ai,j or Ai,j.

### 2.1. Integer Least Squares Model

The GNSS observation equation is [29,30]:(1)y=Aa+Bb+e 
where a is the carrier phase ambiguity parameter, b is the baseline parameter of the component to be estimated, e is the observation noise, y is the carrier phase and pseudorange observation, and A and B are the design matrix.

Using the least squares criterion [31,32], it can be shown that
(2)miny−Aa−BbQy2 a∈ℤm ,b∈ℝn
where ·=·TQy−1·, Qy is the variance covariance matrix of observation y. Considering that the ambiguity parameter is an integer, Equation (2) can be further decomposed as:(3)y−Aa−BbQy2=e^Qy2+a^ −aQa^2+b^a−bQb^|a^2
and
(4)$$\left.\begin{array}{c} \hat{e}=y-A\hat{a}\;-B\hat{b}\\ \hat{b}\left(a\right)=\hat{b}\;-Q_{\hat{b}\hat{a}}Q_{\hat{a}}^{-1}\left(\hat{a}\;-a\right)\\ Q_{\hat{b}|\hat{a}}=Q_{\hat{b}}-Q_{\hat{b}\hat{a}}Q_{\hat{a}}^{-1}Q_{\hat{a}\hat{b}} \end{array}\right\}$$
where $\hat{a}$ is the ambiguity float solution, $\hat{b}$ is the baseline component corresponding to the ambiguity float solution, and $\hat{b}\left(a\right)$ is the baseline component corresponding to the ambiguity fixed solution. Since b in Equation (1) is a real vector, the third term to the right of Equation (3) should be zero. So when $b=\hat{b}\left(a\right)$ and ${\left\|\hat{a}\;-a\right\|}_{Q_{\hat{a}}}^{2}$ takes the minimum value, ${\left\|y-\mathrm{Aa}-\mathrm{Bb}\right\|}_{Q_{y}}^{2}$ takes the minimum value. Therefore, the minimization problem of Equation (2) is transformed into:(5)$$\min{\left\|\hat{a}-a\right\|}_{Q_{\hat{a}}}^{2}=\min{\left(\hat{a}\;-a\right)}^{T}Q_{\hat{a}}^{-1}\left(\hat{a}\;-a\right)\;a\in Z^{m}$$

The Cholesky decomposition of $Q_{\hat{a}}$, namely,
(6)$$Q_{\hat{a}}=G^{T}G$$
where G is the upper triangular matrix.

Substituting Equation (6) into Equation (5) gives
(7)$$\min{\left\|G^{-T}\left(\hat{a}\;-a\right)\right\|}^{2}=\min{\left\|y-G^{-T}a\right\|}^{2}$$
where $y=G^{-T}\hat{a}$ is a constant.

Equation (7) is also known as the nearest vector problem in lattice theory [33]. In order to obtain the integer solution of the ambiguity rapidly, the decorrelation process is usually used to reduce the correlation between the variance components in Q, which improves the efficiency of the search for the ambiguity in Equation (5).

### 2.2. LLL Algorithm Based on QR Decomposition

Let g1 ,g2 ,⋯gn∈ℝn be a set of linearly independent basis vectors and the lattice Lg1 ,g2 ,⋯gn represent the set consisting of all linear combinations of integers of g1 ,g2 ,⋯gn, i.e.,:(8)LG=∑i=1nxigi , xi∈ℤ1≤∀i≤n
where G=g1 ,g2 ,⋯gn is called a set of basis of the lattice, LG is the lattice generated by G, and xi is the combinatorial coefficient of gi.

The classical LLL algorithm implements the reduction on the basis of Gram–Schmidt Orthogonalization (GSO) [11]. Schmidt Orthogonalization is performed on the basis matrix $G=\left[g_{1\;},g_{2\;},\cdots g_{n}\right]$:(9)$$G=G*U=\left[g_{1}^{*},g_{2}^{*},\cdots g_{n}^{*}\right]\left[\begin{array}{cccc} 1 & u_{1,2} & \cdots & u_{1,n}\\ 0 & 1 & \cdots & u_{2,n}\\ 0 & 0 & \ddots & \vdots \\ 0 & 0 & 0 & 1 \end{array}\right]$$

In the formula, $G^{*}=\left[g_{1}^{*},g_{2}^{*},\cdots g_{n}^{*}\right]$ and $g_{i}^{*}=g_{i}-\sum_{j=1}^{i-1}u_{j,i}g_{j}^{*}$, $U=\left[u_{j,i}\right]$ are the unit upper triangular matrix and satisfy $u_{j,i}={\langle g_{i},g_{j}^{*}\rangle /\left\|g_{j}^{*}\right\|}^{2}\;,\;1\leq j<i\leq n$. Matrix $G^{*}$ and U satisfy the following two reduction conditions:(10)$$\left.\begin{array}{c} \left|u_{j,i}\right|\leq \frac{1}{2}\;\;1\leq j<i\leq n\\ \delta {\left\|g_{i-1}^{*}\right\|}^{2}\leq {\left\|g_{i}^{*}\right\|}^{2}+u_{i,i-1}^{2}{\left\|g_{i-1}^{*}\right\|}^{2}\;\;\frac{1}{4}<\delta \leq 1 \end{array}\right\}$$

Call G the LLL reduction basis parameterized by δ. The first equation is called the size reduction and the second equation is the basis vector exchange.

In fact, in order to improve the float accuracy of the lattice basis reduction, the LLL reduction algorithm based on QR decomposition is usually used [34]. The following decomposition is performed on the basis matrix G:(11)$$G=\mathrm{QR}=\left[q_{1},q_{2},\cdots q_{n}\right]\left[\begin{array}{cccc} r_{1,1} & r_{1,2} & \cdots & r_{1,n}\\ 0 & r_{2,2} & \cdots & r_{2,n}\\ 0 & 0 & \ddots & \vdots \\ 0 & 0 & 0 & r_{n,n} \end{array}\right]$$

In the equation, Q is the orthogonal matrix and $q_{i}=g_{i}^{*}/\left\|g_{i}^{*}\right\|$,$R=\left[r_{j,i}\right]$ is the upper triangular matrix and satisfies $u_{j,i}=r_{j,i}/r_{j,j}$ and $\left\|b_{j}^{*}\right\|=r_{j,j}$.

Thus, the reduction condition of Equation (8) can be rewritten as:(12)$$\left.\begin{array}{c} \left|\frac{r_{j,i}}{r_{j,j}}\right|\leq \frac{1}{2}\;\;1\leq j<i\leq n\\ \delta r_{i-1,i-1}^{2}\leq r_{i,i}^{2}+r_{i-1,i}^{2}\;\;\frac{1}{4}<\delta \leq 1 \end{array}\right\}$$

Equation (12) is the reduction condition of the LLL algorithm based on QR decomposition.

In order to satisfy the above reduction conditions, it is usually necessary to construct a transformation matrix for reduction operation.Size reduction: in order to realize the first condition in Equation (10), construct the unimodular matrix $Z_{j,i}=I_{n}-{\left[r_{j,i}/r_{j,j}\right]}_{\mathrm{int}}e_{j}e_{i}^{T}$ (${\left[•\right]}_{\mathrm{int}}$ represents the rounding operator), use its right multiplication by the basis matrix G to realize the size reduction of the corresponding element, and, at the same time, update the upper triangular matrix R.Basis vector exchange (Lovasz condition): if the second condition in Equation (10) is not satisfied, the exchange matrix $P_{i-1,i}$ is constructed to swap the order of $g_{i-1}$ and $g_{i}$, and the matrix R is updated to re-triangularize it.

### 2.3. Improved LLL Algorithm

#### 2.3.1. Householder QR Decomposition Based on Minimum Column Pivoting

The original LLL algorithm performs a QR decomposition of the basis matrix based on GSO. The Householder QR decomposition has lower computational complexity and better numerical stability compared to GSO [35]. Partial LLL reduction algorithms utilize the Householder QR decomposition with minimum column pivoting instead of the regular Householder QR decomposition. In general, the number of basis vector exchanges is a key factor affecting the time consumption of the whole LLL reduction, and it is possible to reduce the number of basis vector exchanges if matrix R of QR decomposition can be made closer to the LLL reduction basis. It can be obtained from Equation (12) that
(13)$$\left(\delta -\frac{1}{4}\right)r_{i-1,i-1}^{2}\leq r_{j,j}^{2}\;\;\frac{1}{4}<\delta \leq 1$$

In order to make it easier for matrix R to satisfy Equation (13), the minimum column pivoting strategy selects the columns that minimize rj,j to be exchanged. In the *j*-th step of the QR decomposition, find column i of Gj:n,j:n which has the shortest length and exchange the i-th column of G with the j-th column. The off-diagonal element Gj+1:n,j is then eliminated by the Householder transformation. By using the minimum column pivoting strategy, the Householder QR expression is:(14)QTGP=R
where P∈ℤn is the exchange matrix and QT=HnHn−1⋯H1 is the product of n Householder transformations.

#### 2.3.2. Partial Size Reduction

It has been theoretically demonstrated in the literature that simple size reduction does not affect the number of candidate points for the ambiguity search, and that basis vector exchange is the real goal of lattice basis reduction to accelerate the ambiguity search. In the LLL algorithm, only the size reduction of the secondary diagonal element is generally required. However, considering the lattice basis reduction efficiency and numerical stability of the algorithm, it is necessary to carry out size reduction for partial non-principal secondary diagonal elements under certain conditions.
(15)$$r_{i-1,i}=r_{i-1,i}-\zeta r_{i-1,i-1}$$
where $\zeta ={\left[r_{i-1,i}/r_{i-1,i-1}\right]}_{\mathrm{int}}$. The size reduction is applied to the non-principal secondary diagonal elements when they satisfy $\left|{\left[r_{i-1,i}/r_{i-1,i-1}\right]}_{\mathrm{int}}\right|\geq 2$, viz:(16)$$r_{k,i}=r_{k,i}-{\left[r_{k,i}/r_{k,k}\right]}_{\mathrm{int}}r_{k,k}\;,\;k=i-2,i-3,\cdots ,1$$

It should be noted that matrix element size reduction is an integer transformation process, which not only reduces the size of the element itself, but also updates the rest of the column vector accordingly.

Givens rotation has better numerical stability than GSO. Therefore, Givens rotation is used for triangularization after the exchange of the basis vector of the PLLL reduction algorithm. Suppose we exchange the k−1 and k columns of R, i.e.,:(17)$$RP_{k-1,k}=\begin{array}{cc} \left[\begin{array}{ccc} R_{1,1} & {\bar{R}}_{1,2} & R_{1,3}\\ & {\tilde{R}}_{2,2} & R_{2,3}\\ & & R_{3,3} \end{array}\right] & \begin{array}{c} {}_{k-2}\\ {}_{2}\\ {}_{n-k} \end{array}\\ \begin{array}{ccc} {}_{k-2} & {}_{2} & {}_{n-k} \end{array} & \end{array}$$
then
$$P_{k-1,k}=\left[\begin{array}{lll} I_{k-2} & & \\ & P & \\ & & I_{n-k} \end{array}\right],\;P=\left[\begin{array}{cc} 0 & 1\\ 1 & 0 \end{array}\right],\;{\tilde{R}}_{2,2}=\left[\begin{array}{cc} r_{k-1,k} & r_{k-1,k-1}\\ r_{k,k} & 0 \end{array}\right],\;{\bar{R}}_{1,2}=\left[\begin{array}{cc} R_{1:k-2,k-1} & R_{1:k-2,k} \end{array}\right].$$

It can be seen that the block matrix ${\tilde{R}}_{2,2}$ is not an upper triangular matrix. Therefore, it is triangularized using Givens rotation. Assuming that the Givens rotation matrix is Γ, we have:(18)$${\bar{R}}_{2,2}:=\Gamma {\tilde{R}}_{2,2}=\left[\begin{array}{cc} c & s\\ -s & c \end{array}\right]\left[\begin{array}{cc} r_{k-1,k} & r_{k-1,k-1}\\ r_{k,k} & 0 \end{array}\right]$$
where
$$c=\frac{r_{k-1,k}}{\sqrt{r_{k-1,k}^{2}+r_{k,k}^{2}}},\;s=\frac{r_{k,k}}{\sqrt{r_{k-1,k}^{2}+r_{k,k}^{2}}}.$$

Therefore, it can be concluded that
$$\Gamma_{k-1,k}RP_{k-1,k}=\bar{R}=\left[\begin{array}{lll} R_{1,1} & {\bar{R}}_{1,2} & R_{1,3}\\ & {\bar{R}}_{2,2} & {\bar{R}}_{2,3}\\ & & R_{3,3} \end{array}\right],\Gamma_{k-1,k}=\left[\begin{array}{lll} I_{k-2} & & \\ & \Gamma & \\ & & I_{n-k} \end{array}\right],{\bar{R}}_{2,3}=\Gamma R_{2,3}.$$

#### 2.3.3. Improvement of the LLL Algorithm

From the PLLL reduction algorithm we note that size reduction is only carried out when the basis vector exchange occurs, and the resulting matrix R is not fully regulated. Therefore, we add an additional size reduction process at the end of the PLLL reduction algorithm and convert R to the LLL reduction matrix. We denote the PLLL algorithm with extra size reduction as PLLLR.

In addition, in the procedure of LLL reduction, it is necessary to detect whether the basis vector satisfies the exchange condition to decide whether it enters the column exchange step or not, and it is obvious that the procedure of LLL reduction can be simplified if the exchange condition of the basis vector is appropriately relaxed in order to reduce the operations, such as the column exchange afterward. Inspired by the literature [36], we replace the Lovasz condition in the LLL reduction with the Siegel condition, and Equation (13) becomes:(19)$$\left(\delta -\frac{1}{2}\right)r_{i-1,i-1}^{2}\leq r_{j,j}^{2}\;\;\frac{3}{4}\leq \delta \leq 1$$

We denote the LLL algorithm based on the Householder QR decomposition as HLLL. The HLLL algorithm is where the basis vector exchange condition is replaced by the Siegel condition as HSLLL, and the PLLL algorithm is where the Siegel condition is used as the basis vector exchange condition as PSLLL. The specific flow of the two improved algorithms is shown in Figure 1.

## 3. Experiments and Results Analysis

In order to verify the effectiveness of the LLL improvement algorithm proposed in this paper in the application of ambiguity resolution, simulation experiments and measured data are used to compare and analyze HLLL, HSLLL, PLLL, PSLLL, and PLLLR, and to evaluate the performance advantages and disadvantages of each algorithm in terms of the extent of reduction basis orthogonality and the quality of reduction basis size reduction. In ambiguity resolution, the searching process of the ambiguity degree adopts the SE-VB strategy which is widely used at present [10]. The experimental environment is a private PC (Intel Core i7-9700 CPU, 2.80 GHz, 16.0 GB of RAM, 64-bit Windows 10 operating system) and the software is MATLAB R2017 a.

### 3.1. Indicators for Evaluating the Quality of the Reduced Basis

In measuring the performance of lattice basis reduction, an orthogonal defect (OD) is usually used to reflect the orthogonality of the basis vector, but it has an obvious disadvantage in that only the OD value is obtained, which is not able to intuitively judge the extent of the orthogonality of the reduced basis [37,38,39]. Therefore, in this paper, the minimum angle θ of the reduced basis vector is used instead of the orthogonality defect to measure the extent of the orthogonality of the reduced basis. Its expression is given as:(20)$$\theta \left(G\right)=\min\left\{\theta_{i,j}\;,\;1\leq i<j\leq n\right\}$$
where
$$\begin{array}{l} \theta_{i,j}=\min\left\{\arccos\left(\rho_{i,j}\right),180{}^{\circ}-\arccos\left(\rho_{i,j}\right)\right\}\\ \rho_{i,j}=\frac{\left(g_{i},g_{j}\right)}{\left\|g_{i}\right\|\left\|g_{j}\right\|} \end{array}$$

By definition, it follows that $0{}^{\circ}\leq \theta \left(G\right)\leq 90{}^{\circ}$. If $\theta \left(G\right)=90{}^{\circ}$, it means that all basis vectors are orthogonal to each other. θ, as an alternative indicator of the extent of orthogonality, can be used to roughly determine the orthogonality of the reduced basis intuitively. And the calculation of θ and OD is based only on the elements of the variance covariance matrix $Q_{\hat{a}}$; it does not increase the computational complexity.

The purpose of the lattice basis reduction is to make the reduced basis as orthogonal as possible and to make the length of the first basis vector as short as possible after the basis vector exchange. Based on this property, the Hermite factor in lattice theory is introduced as another indicator for evaluating the performance of the reduction [40,41], which is defined as:(21)$$\kappa =\frac{\left\|g_{1}\right\|}{{\left[\det\left(Q_{\hat{a}}\right)\right]}^{\frac{1}{2n}}}$$
where $g_{1}$ denotes the first basis vector of the lattice basis G. Obviously, $\det\left(Q_{\hat{a}}\right)$ is a fixed value, then the size of the Hermite factor depends on the length of $g_{1}$. The smaller the value of κ, the shorter the length of the first basis vector after the lattice basis reduction, the more adequate the basis vector exchange, and the better the quality of the reduction, and vice versa.

### 3.2. Simulation Experiment

The random simulation method in the literature [8] is used to construct 5–40 dimensional ambiguity float solution $\hat{a}$ and variance covariance matrix $Q_{\hat{a}}$. Each dimension constructs 100 groups of data, which are processed by HLLL, HSLLL, PLLL, PSLLL, and PLLLR algorithms for lattice basis reduction, respectively. These calculate the average number of basis vector swaps, the average reduction time consumption, and the average number of ambiguity candidate points for the 100 groups of data. The specific construction is as follows:(22)$$\left.\begin{array}{c} \hat{a}=100\times randn\left(n,1\right)\\ Q_{\hat{a}}={\mathrm{LDL}}^{T} \end{array}\right\}$$Scheme 1: L is an upper triangular matrix unit and the upper triangular element $l_{j,i}$ follows the standard normal distribution; $D=diag\left(n^{-1},{\left(n-1\right)}^{-1},\cdots ,1\right)$.Scheme 2: L is a random orthogonal matrix, obtained by the QR decomposition of the random matrix generated by $randn\left(n,n\right)$; $d_{1}=2^{\frac{n}{4}}$, $d_{n}=2^{-\frac{n}{4}}$, $d_{i}\in \left(d_{n},d_{1}\right)$, $D=diag\left(d_{1},\cdots ,d_{i},\cdots ,d_{n}\right)$.

Figure 2 shows the trend of the number of basis vector swaps for the five algorithms in different schemes and dimensions. As seen in Figure 2, the number of basis vector swaps for the five algorithms is positively correlated with the number of dimensions; overall, PSLLL has the fewest number of swaps, and PLLL and PLLLR have the same number of swaps.

By analyzing the results in Figure 2, it can be seen that PLLLR is equivalent to PLLL in terms of the number of basis vector swaps because PLLLR only adds an additional size reduction process, which has no effect on the ordering of the basis vectors, a phenomenon that is in line with the theory. HSLLL and PSLLL simplify the LLL reduction process by relaxing the swap condition of the basis vector, which reduces the number of basis vector swaps.

Figure 3 shows the reduction time consumption for the five algorithms with different schemes and dimensions. It can be intuitively seen from Figure 3 that as the number of dimensions increases, the overall trend of the reduction time consumption for ambiguity is upward, and PSLLL has the smallest reduction times. From Figure 3a, it can be observed that the PLLLR reduction time consumption is lower than the HLLL, except for the 16th dimension. The PSLLL reduction time consumption is lower than the HSLLL (except for the 6th and 11th dimensions). A similar conclusion can be drawn in Figure 3b. The possible reasons for the above special cases are that the reduction time consumption is smaller in the case of lower dimensions and due to the running error of MATLAB.

Figure 4 represents the number of search candidate points for the five algorithms with different schemes and dimensions. As can be seen from Figure 4, the change in the number of search candidate points and dimensions have the same trend overall, that is, the number of candidate points increase with the growth of dimensions. PLLLR and PLLL have the same number of search candidate points, whereas the number of candidate points for the ambiguity of HSLLL and PSLLL is more than HLLL and PLLL in most dimensions compared to the other methods, which indicates that it may be more time consuming in the ambiguity search process.

In analyzing the results of Figure 4, since the simple size reduction does not change the candidate integer vector for the ambiguity search, the number of candidate points for the search of PLLL is equivalent to the search results of the PLLLR algorithm, which is consistent with the theory. Since HSLLL and PSLLL adopt different basis vector exchange conditions from the regular LLL algorithms, the column exchange operation is reduced on the basis vector exchange, thus speeding up the lattice basis reduction procedure. Therefore, the final basis vector lengths obtained are different from those of HLLL and PLLL (the basis vector is obtained by exchanging them in a certain order), which results in a different number of search candidate vectors for HSLLL and PSLLL in different dimensions compared to HLLL and PLLL.

The minimum angle θ and Hermite factor κ between the basis vectors after the reduction of Schemes 1 and 2 using the HLLL, HSLLL, PLLL, PSLLL, and PLLLR algorithms are listed in Table 1 and Table 2, respectively. As can be seen from the average basis vectors of the five algorithms in Table 1 with minimum angles θ, all five algorithms in Scheme 1 have good reduction effects in general. Considering the extent of the orthogonality of the basis vectors, the HSLLL reduction performance is optimal, followed by PLLLR, PLLL, and PSLLL, and HLLL is the worst. Similar conclusions to Table 1 can be drawn from Table 2, but with slight differences in terms of the reduction performance advantages and disadvantages, with PLLLR being the best, followed by PSLLL, HSLLL, and PLLL, and HLLL being the worst. The reason for this difference is that HSLLL and PSLLL are less stable compared to PLLLR. The minimum value of the PSLLL algorithm in Table 1 is 41.3540°, which fluctuates a lot, which means that the orthogonal performance will be poor, whereas both HSLLL and PLLL’s minimum values are greater than 45° and the reduction performance is more stable. Similarly, the same is true for HSLLL in Table 2, which will not be explained here. Combining Table 1 and Table 2, PLLLR is superior in terms of the stability and extent of orthogonality combined.

From the Hermite factor κ of the five algorithms in Table 1 and Table 2, it can be observed that the Hermite factors of PLLLR and PLLL are basically the same, and the relative error is 0.0072%, which is negligible. This indicates that it is difficult to evaluate the performance advantages and disadvantages of the two algorithms from the Hermite factor indicator. There is little difference in the reduction performance between HSLLL and PSLLL, and both outperform the other three algorithms. HSLLL slightly outperforms PSLLL in Scheme 1, while the opposite is true for Scheme 2, which may be related to the type and randomness of the reduced basis. HLLL has the worst reduction performance.

### 3.3. Measured Experiment 1

To further validate the effectiveness of the algorithm and the reduction effect, using the GPS dual-frequency observation data from the US CORS station LWES with DSTR on 15 March 2023 (DOY-074) for 2778 epochs, the baseline length is 7.79 km and the sampling interval is 30 s. The ambiguity dilution of precision (ADOP) is usually used to evaluate the accuracy of the ambiguity resolution [42]. Figure 5 shows the variation trend of the ambiguity dimension and ADOP in DOY-074. It can be observed from Figure 5 that the ambiguity dimension of DOY-074 ranges from 12 to 22 dimensions, and the ADOP value is all less than 0.1. The dimensions of the first 200 epochs are about 20, and the ADOP value is all less than 0.06. Therefore, in this paper, we select the data of the first 200 epochs to verify the effectiveness and reduction performance of the improved algorithm.

Figure 6 shows the cumulative distribution functions of the number of basis vector swaps and reduction time consumption for the first 200 epochs of the five algorithms. It can be seen from Figure 6a that PLLL and PLLLR have the same number of basis vector swaps, and PSLLL has the smallest number of basis vector swaps, followed by HSLLL. This is consistent with the conclusion of the simulation experiments in Section 3.2. From Figure 6b, it can be observed that the reduction time consumption of PSLLL and HSLLL is significantly less than the other three algorithms, and PLLLR consumes slightly more reduction time than PLLL due to the extra added size reduction, which is in line with the theory in Section 2.3. The five algorithms in descending order of reduction efficiency are PSLLL, HSLLL, PLLL, PLLLR, and HLLL.

Figure 7 shows the variation of the number of ambiguity candidate points for the five reduction algorithms in the first 200 epochs, from which it can be seen that the number of ambiguity candidate points is exactly the same for PLLL and PLLLR, which is consistent with the conclusion of the simulation experiments, and will not be explained here. HLLL, HSLLL, and PSLLL have different numbers of candidate points for ambiguity, and the differences between HLLL and HSLLL can be clearly seen in the figure, while the overall trend of PSLLL is in line with PLLL and PLLLR.

Table 3 shows the statistical results for the five algorithmic basis vectors’ minimum angles (deg) and Hermite factors in the first 200 epochs. As seen in Table 3, the average basis vector minimum angle of all five algorithms is greater than 45°, and the PLLLR has the best reduction performance. From the Hermite factors κ of the five algorithms, it can be observed that the order of the reduction performance is consistent with Scheme 2 in the simulation experiments, PLLLR and PLLL have the same Hermite factor, and PSLLL outperforms several other methods.

Table 4 represents the solution time consumption (reduction time consumption, search time consumption, and total time consumption) of the five algorithms for the first 200 epochs, from which it can be observed that the five algorithms have the highest overall efficiency in the order of PSLLL, HSLLL, PLLLR, PLLL, and HLLL. The PSLLL has the highest overall efficiency. The PLLLR algorithm has the highest search efficiency by further size reduction and has the best stability, which is favorable for improving the search efficiency of ambiguity.

### 3.4. Measured Experiment 2

In order to further verify the reduction performance of the algorithm in the case of multiple GNSS systems and higher dimensionality, the simulated railroad track measured GPS/BDS data of 1210 epochs from Southwest Jiaotong University on 16 August 2023 (DOY-228) are selected, with a baseline length of 9.80 m and a sampling interval of 1 s. Figure 8 shows the trend plot of the ambiguity dimension and ADOP for 1210 epochs. From the figure, it can be seen that the number of ambiguity dimension is greater than 36 and the value of ADOP is less than 0.07. Therefore, the accuracy of the float solution of the ambiguity is better.

Figure 9 shows the cumulative distribution functions of the number of basis vector swaps and the reduction time consumption for the five algorithms, from which it can be observed that PLLL and PLLLR have the same number of basis vector swaps. As the number of ambiguity dimensions is close to 40, the variation of the reduction time consumption of PSLLL and HSLLL is small, and the reduction time consumption of PSLLL and HSLLL is significantly smaller than the other three algorithms. The reduction time consumption of PLLLR adding extra size reduction does not increase significantly compared with PLLL, which is due to the extra size reduction with low complexity, and the time consumption is basically negligible. There is no difference in the trend of the number of ambiguity candidate points of the five algorithms, which will not be shown here.

Table 5 shows the basis vector minimum angle (deg) and Hermite factor of the five algorithms. It can be seen that the average basis vector minimum angle θ of the five algorithms is greater than 45°, and all of them have good reduction effects. The minimum value of θ of the PLLLR algorithm is greater than 45°, which indicates that PLLLR has the best robustness in avoiding the reduced basis of poor orthogonality. The Hermite factor κ of PLLLR and PLLL is basically the same, and the relative error is negligible. The superiority of the PLLLR and PLLL algorithms cannot be judged from the Hermite factor κ alone.

Table 6 shows the solution time consumption of the five algorithms (reduction time consumption, search time consumption, and total time consumption), from which the conclusions are consistent with those of Table 4 in Measured Experiment 1, and will not be repeated here. Figure 10 illustrates the cumulative distribution functions of the total time consumed for the two measured experiments, from which it can be seen that the HSLLL, PSLLL, and PLLLR algorithms outperform the HLLL and PLLL. The difference is that it is not possible to ascertain the performance of the HSLLL and PLLLR algorithms from Measured Experiment 1, whereas Measured Experiment 2 clearly shows that the efficiency of the HSLLL outperforms that of the PLLLR. The possible reasons for this difference are related to the number of ambiguity dimensions and MATLAB running errors.

In order to illustrate the performance difference between HLLL, HSLLL, PLLL, PSLLL, and PLLLR more clearly, we compare the speed, stability, and computational complexity of the five algorithms, and the results are shown in Table 7.

## 4. Discussion

The classical LLL algorithm is based on the QR decomposition of the basis matrix by GSO. The computational complexity of GSO is $2n^{3}$, while the Householder QR decomposition does not require the formation of orthogonality factor Q during the reduction process, and the computational complexity is 43n3. In addition, the GSO method has poor numerical properties because there is usually a severe loss of orthogonality in the computation of the orthogonality factor Q. Therefore, in this paper, the Householder QR decomposition is utilized instead of the conventional GSO decomposition, which has lower computational complexity and better numerical stability. We propose corresponding improved algorithms based on the HLLL and PLLL algorithms, and verify the validity of the methods and the performance of the reduction through simulation and measured experiments. Figure 2, Figure 6a, and Figure 9a represent the number of basis vector swaps for the simulation and measured experiments, from which it can be seen that PSLLL has the smallest number of basis vector swaps, and PLLLR has the same number of basis vector swaps as PLLL. From the reduction time consumption of different algorithms in Figure 3, Figure 6b, and Figure 9b, it can be seen that the reduction time consumption of PSLLL and HSLLL is less than the other algorithms, whereas PLLLR makes the reduction time slightly higher than PLLL due to the extra size reduction. Combined with the cumulative distribution functions of the total time consumption (the sum of the reduction time and the search time) of the two measured experiments in Figure 10, it can be seen that although the PLLLR reduction time consumption is slightly higher than that of PLLL, the further size reduction of the R-matrix after exchanging the basis vectors greatly shortens the search time, which improves the overall efficiency of the ambiguity resolution. Figure 4 and Figure 7 analyze the five algorithms from the comparison of the number of ambiguity candidate points. The number of ambiguity candidate points is exactly the same for PLLLR and PLLL, while the other algorithms are slightly different. Table 4 and Table 6 show the solution time consumption of the five algorithms, where PSLLL has the fastest reduction efficiency and PLLLR has the best stability. Table 7 summarizes the performance differences of the five algorithms.

There is a close correlation between lattice basis orthogonality and basis vector length, and the ambiguity lattice basis reduction is precise to better solve the CVP on the lattice. In measuring the performance of the lattice basis reduction, the extent of orthogonality between the reduced basis vector is not intuitively determined due to the orthogonal defect indicator. We introduce the minimum angles θ among the reduced basis vectors as an alternative to overcome the drawbacks of the orthogonal defect. It can be found through Equation (20) that a good lattice basis reduction algorithm should ensure that θ is greater than 45°. As can be seen from the minimum angles among the basis vectors in Table 1, Table 2 and Table 3 and Table 5, the minimum values of θ for the PLLLR algorithm are all greater than 45°, which suggests that it is the most robust in terms of avoiding a reduced basis with poor orthogonality. The Hermite factors can well reflect the property of the first basis vector of the lattice basis reduction, that is, whether the length of the first orthogonal basis vector is short enough. From the Hermite factors statistics in Table 1, Table 2 and Table 3 and Table 5, it can be observed that HSLLL and PSLLL outperform the other three algorithms in terms of reduction performance. PLLLR and PLLL have basically the same Hermite factors with negligible relative errors. It is difficult to evaluate their performance advantages or disadvantages from the Hermite factors.

## 5. Conclusions

In this paper, for the characteristics of the high dimensionality and high accuracy of ambiguity resolution, based on analyzing the LLL reduction algorithm, we introduce the minimum column pivoting Householder QR decomposition, partial size reduction, and the relaxation of the basis vector exchange condition to improve the regular LLL algorithm. In order to visualize the extent of the orthogonality of the basis vectors, the minimum angle of the basis vectors is used to replace the conventional degree of orthogonality defects, and the quality of the reduced basis size reduction is evaluated by the Hermite factor. Based on the simulation and measured data to verify the effectiveness of the improved algorithm and the reduction effect, the experimental results show that the improved algorithm effectively reduces the size reduction and the number of basis vectors exchanged in the process of lattice basis reduction, and can obtain a better reduction effect, which significantly improves the reduction performance of the LLL algorithm. HSLLL and PSLLL have a better reduction effect, but are poor in the stability performance of lattice base reduction. The PLLLR algorithm loses a small amount of reduction time, but improves the search efficiency of the ambiguity, which effectively improves the overall efficiency of the ambiguity resolution.

## Figures and Tables

**Figure 1 sensors-23-08568-f001:**
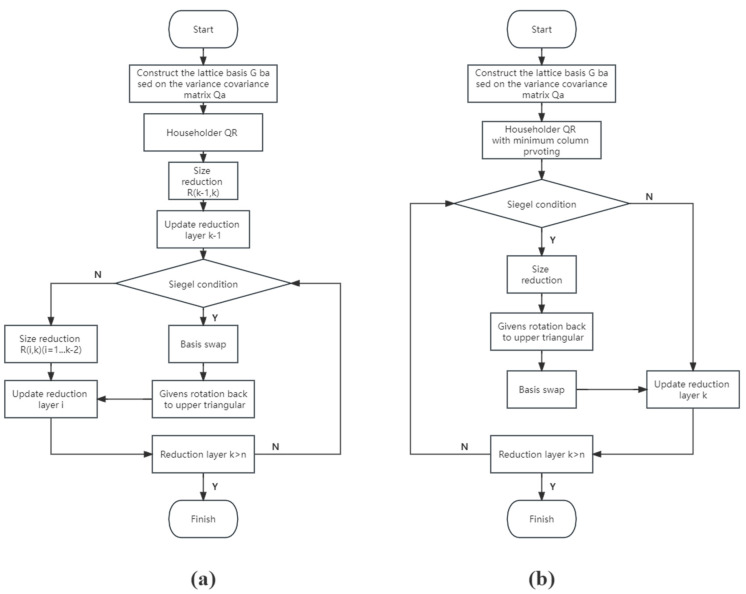
(**a**) Flow chart of HSLLL algorithm; (**b**) Flow chart of PSLLL algorithm.

**Figure 2 sensors-23-08568-f002:**
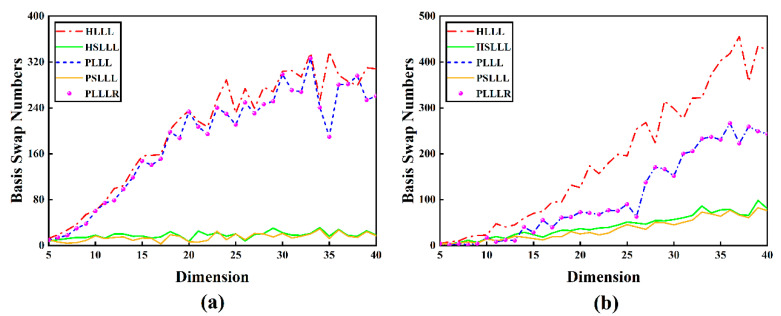
(**a**) Number of basis vector swaps in different dimensions of Scheme 1; (**b**) Number of basis vector swaps in different dimensions of Scheme 2.

**Figure 3 sensors-23-08568-f003:**
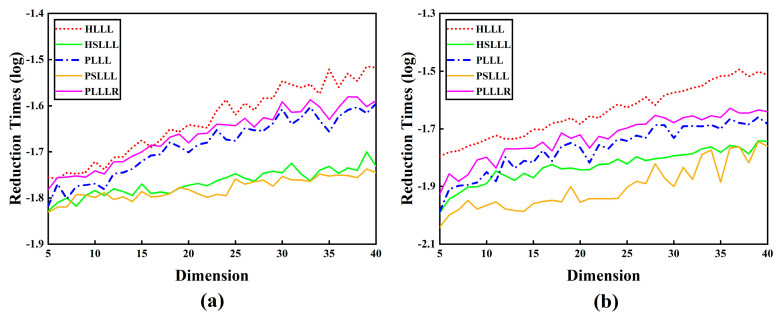
(**a**) Reduction time consumption in different dimensions of Scheme 1; (**b**) Reduction time consumption in different dimensions of Scheme 2.

**Figure 4 sensors-23-08568-f004:**
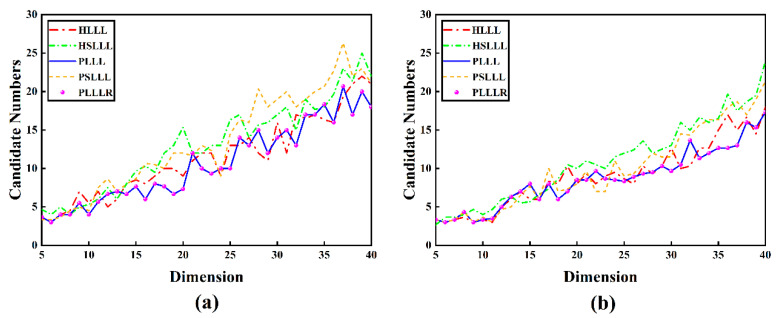
(**a**) The number of search candidate points in different dimensions of Scheme 1; (**b**) The number of search candidate points in different dimensions of Scheme 2.

**Figure 5 sensors-23-08568-f005:**
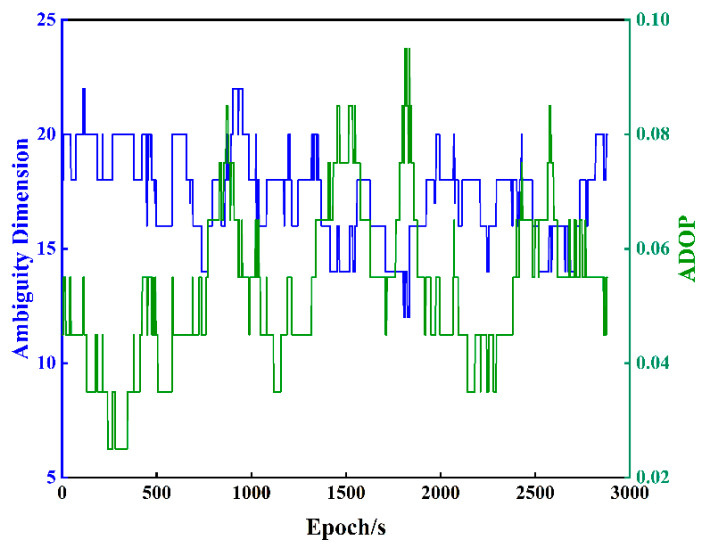
Ambiguity dimensions and ADOP values for DOY-074.

**Figure 6 sensors-23-08568-f006:**
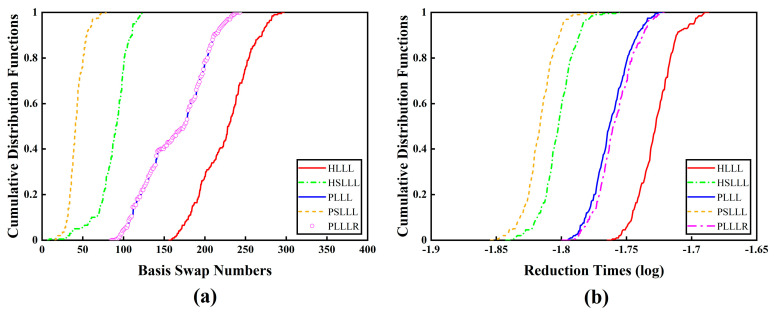
(**a**) Plot of the cumulative distribution functions of the number of basis vector swaps for the five algorithms in the first 200 epochs; (**b**) Plot of the cumulative distribution functions of the reduction time consumption for the five algorithms in the first 200 epochs.

**Figure 7 sensors-23-08568-f007:**
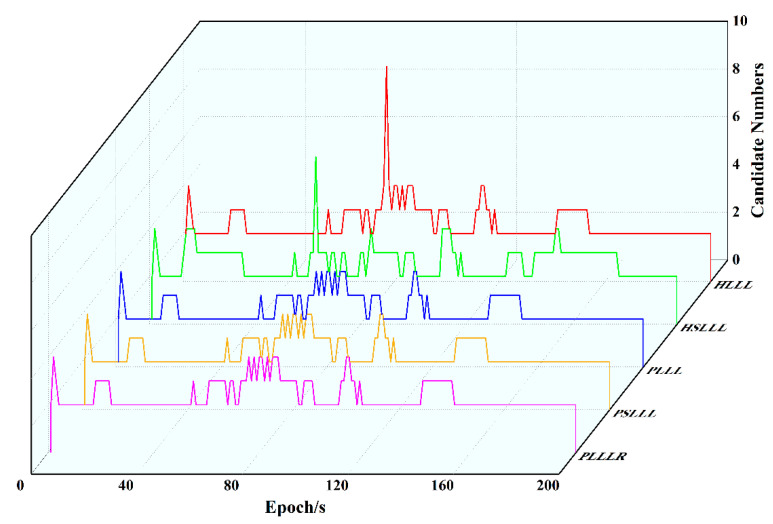
Number of ambiguity candidate points for the five algorithms in the first 200 epochs.

**Figure 8 sensors-23-08568-f008:**
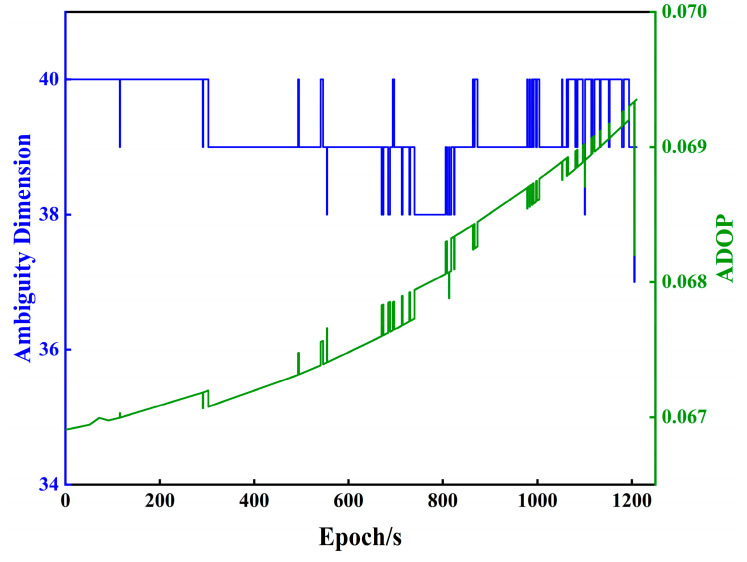
Trend of ambiguity dimension and ADOP for 1210 epochs of DOY-228.

**Figure 9 sensors-23-08568-f009:**
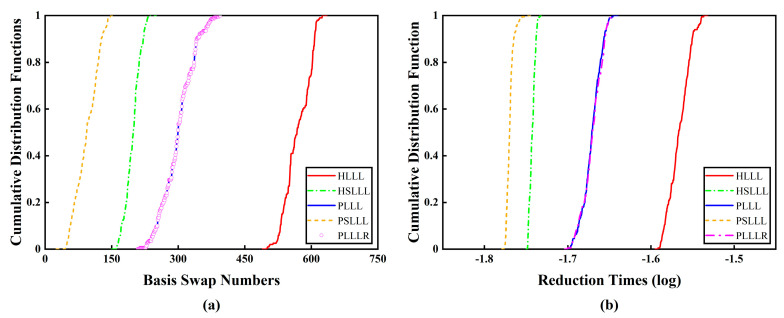
(**a**) Plot of the cumulative distribution functions of the number of basis vector swaps for the five algorithms; (**b**) Plot of the cumulative distribution functions of the reduction time consumption for the five algorithms.

**Figure 10 sensors-23-08568-f010:**
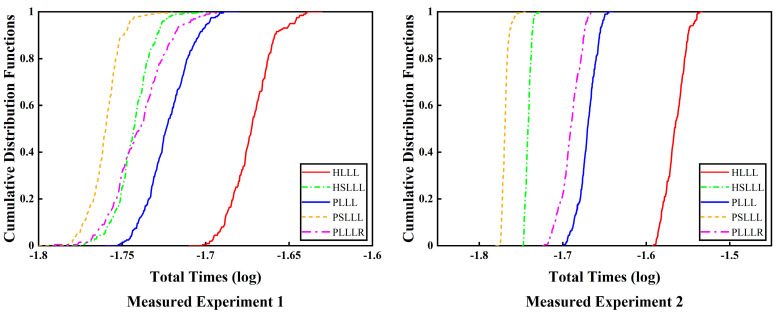
Plot of the cumulative distribution functions of the total time consuming of the five algorithms for the two measured experiments.

**Table 1 sensors-23-08568-t001:** Basis vector minimum angle (deg) and Hermite factor for the five algorithms of Scheme 1.

Methods	HLLL		HSLLL		PLLL		PSLLL		PLLLR	
	θ	κ	θ	κ	θ	κ	θ	κ	θ	κ
Max	65.0848	0.7704	72.4634	0.6544	68.6206	0.7537	65.6931	0.6544	68.6206	0.7537
Min	41.1187	0.6279	46.2028	0.6277	43.5084	0.6277	41.3540	0.6277	45.6056	0.6277
Mean	51.2901	0.6622	58.7290	0.6355	54.7261	0.6599	52.5564	0.6361	57.1343	0.6598

**Table 2 sensors-23-08568-t002:** Basis vector minimum angle (deg) and Hermite factor for the five algorithms of Scheme 2.

Methods	HLLL		HSLLL		PLLL		PSLLL		PLLLR	
	θ	κ	θ	κ	θ	κ	θ	κ	θ	κ
Max	67.2323	1.0223	67.5014	0.9893	67.3050	0.9893	67.3050	0.9893	67.3050	0.9893
Min	39.3641	0.7985	44.3996	0.4801	45.4427	0.6445	47.4158	0.4801	47.6255	0.6445
Mean	50.0438	0.8652	55.4014	0.7804	54.7020	0.8237	57.0028	0.7705	57.7065	0.8237

**Table 3 sensors-23-08568-t003:** Basis vector minimum angle (deg) and Hermite factor for five algorithms of Measured Experiment 1.

Methods	HLLL		HSLLL		PLLL		PSLLL		PLLLR	
	θ	κ	θ	κ	θ	κ	θ	κ	θ	κ
Max	52.0347	2.1952	55.8564	2.0917	53.4141	2.1952	52.1806	2.0917	56.5125	2.1952
Min	41.0043	1.1528	44.0579	1.1527	42.1093	1.1528	41.4327	1.1527	45.8418	1.1528
Mean	45.5869	1.7399	50.3149	1.7338	47.6742	1.7383	46.8228	1.7331	50.7791	1.7383

**Table 4 sensors-23-08568-t004:** Statistical results of five algorithms’ resolution times for Measured Experiment 1 (ms).

Time	Methods	HLLL	HSLLL	PLLL	PSLLL	PLLLR
Reduction	mean	18.7527	15.7789	17.2777	15.3097	17.3245
	max	20.4329	18.8352	18.6567	18.9781	18.4936
Search	mean	2.5271	2.2992	1.6384	2.0732	0.9467
	max	4.9388	4.3730	2.7543	4.7754	2.3275
Total	mean	21.2798	18.0781	18.9161	17.3829	18.2712
	max	24.8507	22.8804	20.8045	22.3506	20.5054

**Table 5 sensors-23-08568-t005:** Basis vector minimum angle (deg) and Hermite factor for five algorithms of Measured Experiment 2.

Methods	HLLL		HSLLL		PLLL		PSLLL		PLLLR	
	θ	κ	θ	κ	θ	κ	θ	κ	θ	κ
Max	53.1254	2.1676	55.1371	2.0435	56.4574	2.1676	55.6064	2.0434	58.1738	2.1675
Min	42.2783	1.1366	43.4854	1.1364	44.0098	1.1365	43.7539	1.1364	46.1105	1.1365
Mean	46.0438	1.7013	48.5130	1.6954	48.1327	1.6977	49.6103	1.6912	51.0883	1.6977

**Table 6 sensors-23-08568-t006:** Statistical results of five algorithms’ resolution times for Measured Experiment 2 (ms).

Time	Methods	HLLL	HSLLL	PLLL	PSLLL	PLLLR
Reduction	mean	27.1813	18.0880	21.3378	17.8412	21.3639
	max	29.0811	28.6279	22.6296	27.8739	22.5280
Search	mean	3.8941	3.3249	2.8544	2.9592	1.7886
	max	6.1841	5.2118	3.2176	5.1017	2.8402
Total	mean	31.0754	21.4129	24.1922	20.8004	23.1525
	max	34.2219	30.5801	25.6345	29.6266	24.5172

**Table 7 sensors-23-08568-t007:** Comparison of the five algorithms.

Method	HLLL	HSLLL	PLLL	PSLLL	PLLLR
Reduction speed	Slow	Faster	Fast	Fastest	Faster
Search speed	Slow	Fast	Faster	Faster	Fastest
Stability	Good	Good in most cases	Better	Good in most cases	Best
Complexity	$ο\left(\frac{4}{3}n^{3}+n^{5}+n^{4}\log\frac{\alpha }{\beta }\right)$ *	Same as HLLL	Same as HLLL	Same as HLLL	$ο\left(\frac{7}{3}n^{3}+n^{5}+n^{4}\log\frac{\alpha }{\beta }\right)$ *

* $\alpha =\max\left\|g_{i}\right\|$ and $\beta =\min\left\|G^{-T}a\right\|$.

## Data Availability

The dataset supporting this research can be found at the NOAA’s National Geodetic Survey (NGS).

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
