# Peer review of "Improved GNSS Ambiguity Fast Estimation Reduction Algorithm"

_sensors, 2023, doi:10.3390/s23208568_

Round 1

Reviewer 1 Report

In this contribution, the author proposed a method for GPS ambiguity reduction. The review, however, has the following comments:

- Based on the literature review, the proposed method has been already investigated in many previous studies. What is the author’s contribution?

- The authors used a simulated data although GPS/GNSS datasets are available from several sources. Why? Even the used simulated dataset was very old!

- Why did the author use this simulated dataset particularly?

- For real data, only one day dataset over one reference station was used.

- The obtained results needs more statistical analysis than the used with more illustrative figures.

- The mainly used abbreviations were not explained (i.e., HLLL, HSLLL, PLLL, PSLLL and  PLLLR).  

- Is the proposed method applicable for other GNSS systems? Although GNSS was mentioned in the title and GPS is only used.  

- Lines 94-99: should be moved to mathematical model section NOT in the Introduction section.

- In Experiments and results analysis section: it will be better if a table is added in order to summarize the main differences between HLLL, HSLLL, PLLL, PSLLL and PLLLR algorithms.

So, the reviewer see that the paper is not suitable for publication in Sensor journal.  

Reviewer 2 Report

1. On page 9, lines 307~316, there are some errors in the description of Figure 3. The original description is "From Fig. 3(a), it can be observed that the PLLLR reduction time consuming is lower than the HLLL."  in fact this is not the case, and it is difficult to tell from Figure 3.

2. In Figure 7, the curves of PLLL, PSLLL, PLLLR are all consistent. Therefore, on page 13, the description of lines 400-401 is problematic and needs to be confirmed and corrected again.

Reviewer 3 Report

In this article, the author proposes an improved GNSS ambiguity fast estimation reduction algorithm ,which can reduce the number of basis vector exchanges and the reduction time consuming. The experimental results also confirmed this viewpoint. But, I am more concerned about the effectiveness of improved GNSS ambiguity fast estimation and reduction algorithm in fixing success rates wtih fixed ambiguity. And what are the advantages and disadvantages compared to traditional methods?

Round 2

Reviewer 1 Report

None

Reviewer 3 Report

I have no another questions.